

# Downslope windstorm study in the Isthmus of Tehuantepec using WRF high-resolution simulations

Miguel A. Prósper[1], Ian Sosa Tinoco[2], Carlos Otero-Casal[1,3], and Gonzalo Miguez-Macho[1]

[1]Universidade de Santiago de Compostela, Spain.
[2]Technical Institute of Sonora, Sonora, Mexico.
[3]MeteoGalicia, Xunta de Galicia, Santiago de Compostela, Spain

**Correspondence:** Miguel A. Prósper (prosper.miguelangel@gmail.com)

**Abstract.**

Tehuantepecers or Tehuanos are extreme winds produced in the Isthmus of Tehuantepec, blowing south through the Chivela pass, the mountain gap across the isthmus, from the Gulf of Mexico into the Pacific Ocean. They are the result of the complex interaction between large-scale meteorological conditions and local orographic forcings around Chivela pass and occur mainly

in winter months, due to cold air damming in the wake of cold fronts that reach as far south. They can generate localized extreme events, such as downslope windstorms and hydraulic jumps, strong turbulent flows that have a direct effect on the Pacific side of the isthmus and the Gulf of Tehuantepec.

This study focuses on investigating these phenomena using high horizontal and vertical resolution WRF (Weather Research and Forecasting) model simulations. In particular, we employ a four nested grid configuration, with up to 444 m horizontal

spacing in the innermost domain and 70 hybrid-sigma vertical levels, 8 of which lie within the first 200 m above ground. We select one 36 hour period in December 2013, when favorable conditions for a strong gap wind situation were observed. The high-resolution WRF experiment reveals a significant fine-scale structure in the strong Tehuano wind flow, beyond the well known gap jet. Depending on the Froude number upstream the topographic barrier, different downslope windstorm conditions and hydraulic jumps with rotor circulations develop simultaneously at different locations east of Chivela pass with varied

crest height. A comparison with observations suggests that the model accurately represents the spatially heterogeneous intense downslope windstorm and the formation of mountain wave clouds for several hours, with low errors in wind speed, wind direction, and temperature.

*Copyright statement.* TEXT

# 1   Introduction

The Isthmus of Tehuantepec, in the Mexican state of Oaxaca, is the narrowest stretch of land separating the Gulf of Mexico from the Pacific Ocean. The Sierra Madre mountains cross the isthmus from east to west, leaving, however, a pronounced gap in the middle (Chivela Pass), coinciding with the point of shortest distance between the two sea masses, of only 200 km. The



elevation of the Chivela pass is 224 m, whereas mountain peaks in the side sierras reach 2000 m, creating ideal conditions to generate a powerful wind corridor (Romero-Centeno et al., 2003). In winter, cold high-pressure systems originating in North America move over the Gulf of Mexico in the wake of south-reaching cold fronts, and large pressure differences develop across the isthmus between the bay of Campeche and the Gulf of Tehuantepec, on the Pacific side. This pressure gradient results in

a northerly wind situation, in which the flow is accelerated southward by cold air damming, traveling through Chivela Pass to finally blow violently outward into the Pacific Ocean. In the Gulf of Tehuantepec, the strong sea surface wind stress generates intense upwelling and vertical mixing in the upper ocean (Hong et al., 2018). These powerful mountain gap winds are called Tehuantepecers or Tehuanos, and have been the focus of several previous studies (Brennan et al., 2010; McCreary et al., 1989; Jaramillo and Borja, 2004) detailing the general setting, drivers and dynamics (Steenburgh et al., 1998) of strong wind

situations in the isthmus. The largest number of these events tend to occur in December, with a mean duration of 48h (Brennan et al., 2010). Little knowledge exists, however, on the fine-scale structure of the Tehuantepecer flow, which is prone to result in downslope windstorms (DSWS hereafter) potentially producing severe turbulent phenomena such as rotors and hydraulic jumps (Durran, 1986a; Sheridan and Vosper, 2006) on the Pacific side of the isthmus, to the lee of local orography, particularly during the cold season. There is evidence from observations and earlier numerical studies (Steenburgh et al., 1998) that the low

elevation topography of Chivela pass can excite mountain waves, and also that these are not only restricted to the pass itself but extend into the much higher mountain crests to the west and especially to the east, as the cold air pool is often thick enough to surpass them. The ability to understand and forecast these events is very relevant, since the Isthmus has been an important development site for wind farms since the 2000's (Coldwell et al., 2017). Currently this region allocates 76.8% of the wind power capacity installed in Mexico, with approximately 2360 MW (Baxter et al., 2017), which is expected to double to 5076

MW by 2020 (AMDEE ,2018). In addition, several accidents related to the strong winds are reported by the Oaxaca´s Civil Protection Commission (Santiago, 2018; Hernández, 2018; Televisa, 2018; Rodríguez, 2018) every year during some Tehuano occurrences.

The main goal of the present work is to study the variability of flow behavior in Tehuano wind episodes across the isthmus of Tehuantepec, depending on topographic barrier height and thermodynamic conditions of the air mass, using high-resolution

simulations with the Weather Research and Forecast (WRF) model. Many studies have successfully employed WRF to analyze this kind of mountain-flow events in other parts of the world. In the US, for example, B. Pokharel et al. Pokharel et al. (2017b) study a DSWS and hydraulic jumps to the lee of the Medicine Bow Mountains in southeast Wyoming. Another DSWS in this same area is also investigated by Grubišić et al. (2015) with WRF. Other studies such as Cao and Fovell (2016), Pokharel et al. (2017a), Prtenjak and Belusic (2009), Jung-Hoon and Chung (2006), Ágústsson and Ólafsson (2014), Priestley et al. (2017)

also use WRF at high resolution to analyze mountain wave flows in different locations. To the best of our knowledge, there is not any previous work that studies in detail lee wave phenomena in the Mexican state of Oaxaca.

The high-resolution WRF simulations employed in the present study allow us to obtain a more complete knowledge of these events, from the synoptic scale to the microscale, focusing on the downslope winds and the hydraulic jumps that develop along the Chivela pass and the neighboring mountain ranges. The article is organized as follows: in section II the methodology is

explained in detail, from the climatology of the region to the model configuration. In section III, the primary results obtained are



shown, divided by the synoptic-mesoscale situation, the upstream-downstream structure of the phenomena and the microscale situation. Finally, in section IV the conclusions reached are discussed.

## 1.1 Mountain wave phenomena, hydraulic analog and relation with the Froude number

Downslope windstorms result from the intense flow acceleration occurring on the lee slope of a mountain under certain cross-
barrier wind conditions. They resemble the behavior of water flow in an open channel when encountering an obstacle, such as when a relatively slow river increases its speed when flowing in a thin layer over a mill's dam or over a rock. As in the case of the river, downslope windstorms often end abruptly with a return to the state upstream of the obstacle through a turbulent hydraulic jump somewhere downstream. The similarity in both fluid's behavior suggests that the physical processes behind are also alike, and that shallow-water theory could be applied somehow to the atmosphere (Long, 1953). However, the
complexity of the unbounded atmospheric flow, without a free surface, makes it difficult to make the analog so simple, because gravity waves in the atmosphere propagate vertically in addition to horizontally as in shallow water, and non-linear effects are important. In shallow water theory, the Froude number (Fr), which is the ratio of the mean speed to the intrinsic gravity wave phase speed in the fluid, determines its behavior when encountering the obstacle, depending on whether gravity is balanced mostly by acceleration (Fr>1) or pressure gradient forces (Fr<1). Hydraulic jumps and significant flow acceleration to the lee
of the obstacle occur when the fluid transitions from a subcritical (Fr <1) to supercritical (Fr>1) regime at the top of the barrier. In this case the fluid thins and speeds up while approaching the crest like in subcritical conditions, but instead of returning to the prior state, it keeps thinning and accelerating on the way down as a supercritical flow (Durran, 1986b). Fr must, thus, be around 1 from the start, or in other words, the relation among flow speed, flow depth, which determines gravity wave speed, and obstacle height should be about right.

Considerable observational and numerical experiments have been developed to elucidate the dynamics of atmospheric mountain lee flows (see Durran (1990) for a review). The current view is that in the atmosphere, downslope windstorms are observed when stable air at low levels flows, similarly to shallow water, as like having a free surface somewhere above the obstacle preventing vertical energy dissipation. This occurs when gravity waves do not propagate vertically and break because of the presence of a critical level (Durran and Klemp, 1987) or due to overturning for having too large amplitude (Clark and Peltier,
1977), or when without wave breaking, there is an interface separating highly stable lower layers from less stable air above (Durran, 1986b; Klemp and Durran, 1987; Bacmeister and Pierrehumbert, 1988). Wave breaking creates a well-mixed region to the lee of the obstacle that induces flow separation; the generation of a dividing streamline between undisturbed flow above and trapped energy and flow analogous to hydraulics in the lower surficial branch (Smith, 1985b). A highly stable lower layer topped by less stable air results in reflecting and decaying waves aloft, enhanced non-linear effects, and the atmosphere also
flowing like a two-layer fluid and behaving like shallow water when encountering the barrier. The height of the aforementioned dividing streamline or the thickness of the highly stable layer at low levels play a similar role to that of the depth of shallow water for atmospheric flows over an obstacle. In either case, strong flow acceleration on the lee slope and hydraulic jumps are produced when there is a transition from a subcritical like regime on the windward side of the mountain to supercritical like conditions on the lee, as in hydraulic theory. Increasing speeds upslope are due to pressure gradient forces in the standing





gravity wave generated by the mountain dominating over gravity (subcritical flow). If the barrier is high enough, velocities at the crest may become so large that lee side gravity-wave induced pressure gradient forces are unable to decelerate the flow, gravity dominates (supercritical flow) and speeds further increase on the way down attaining very high values (Durran, 1990).

It is not straightforward to define a Froude number to determine the regime of atmospheric flows as it is for shallow water,
because there is no clear analog to flow depth controlling gravity wave speed and pressure perturbations. Key factors determining wave properties and flow characteristics in this case are stratification, wind speed and barrier height, or rather, the relative values among them. A quantity that combines these three variables and is referred to as Froude number for shallow atmospheric flows over mountains by many authors, is the following (Smith, 1989):

$$Fr = \frac{U}{NH} \tag{1}$$

where $U$ is the flow speed, $N$, the Brunt-Väisäla frequency (Equation (2)), and $H$, the mountain height.

$$N = \sqrt{\frac{g}{\theta_0} \frac{d\theta}{dz}} \tag{2}$$

with $g$ the acceleration of gravity, $d\theta/dt$, the potential temperature gradient in the stable layer, and $\theta_i$, the potential temperature at the base of this layer. To avoid confusion with the classical Froude number, the inverse of Fr defined as above is often used instead and called the non-dimensional mountain height. Fr is a measure of flow deceleration and stagnation upwind
from the mountain (Baines, 1987). It can also be regarded as an estimate of nonlinearity. When Fr«1 there are significant nonlinear effects and blocking, whereas for Fr»1 the opposite occurs (Smolarkiewicz and Rotunno, 1989). Fr around 1 indicates a transitional regime between the two states and favorable conditions for the formation of DSWS and hydraulic jumps. The flow dynamics related to a DSWSs and HJs in the atmosphere is complex (Clark and Peltier, 1984), and there are different Fr calculations that take into account further factors (Sheridan and Vosper, 2006; Smith, 1989, 1985a). However, due to the flow
type for the period studied, a steady wind due south, and a fairly constant stratification with height in the lower layers, Fr in (Eq3) can be considered a good indicator of the flow characteristics in the considered area, especially in HJ cases.

## 2   Methodology

### 2.1   WRF Configuration

We use the Advanced Research WRF (ARW) model (Skamarock et al., 2008) version 3.9 (WRFV3.9) to perform the simula-
tions. Based on a fully compressible and non-hydrostatic dynamic core, WRFV3.9 is a limited-area mesoscale and microscale model, with a terrain-following hydrostatic-pressure vertical coordinate, designed for operational forecasting, as well as research. For the experiments, we employ a nested domain configuration, in order to achieve sufficiently high resolution in the innermost grids to capture the small scale structure of the flow, while reproducing the synoptic phenomenology conducive to local DSWS in the parent one (Figure 1a).

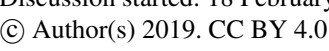



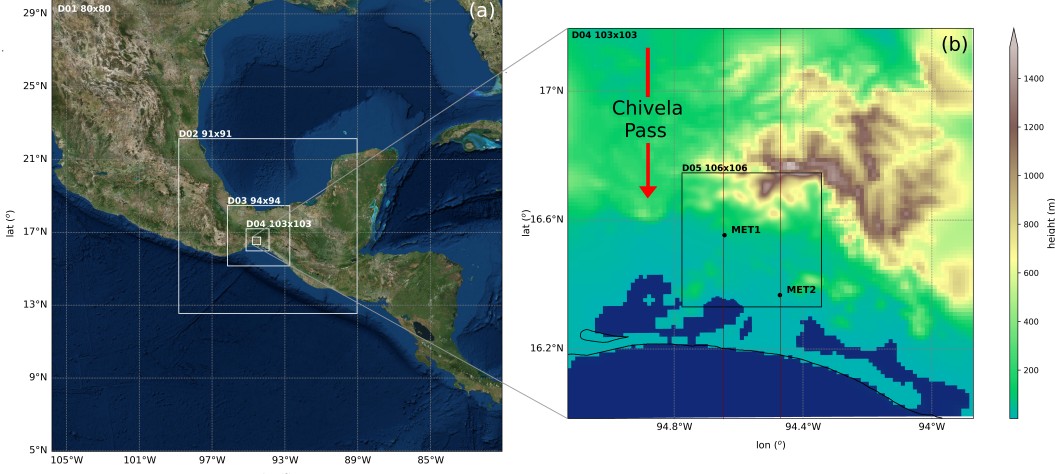

**Figure 1.** WRF nested domain configuration. (a) Coarser three domains with their number of grid points. (b) Higher resolution domains (d04 and d05), both with their respective topographies (m above sea level). MET1 and MET2 are the locations of the meteorological stations used as validation points. The two red lines represent the vertical cross-sections shown in Figures 3 and 5.

The domain's configuration meets the requirements recommended by Warner (2011), including a parent (d01) and four nested grids (d02, d03, d04 and d05) (Figure1a-b) one-way interacting. D01 is centered at 17.91 N and 93.44 W (Figure2c) with 80 x 80 grid points of 36 km of horizontal resolution. The horizontal resolutions of d02, d03, d04, and d05 are: 12 km (91x91 grid points), 4 km (94x94 grid points), 1.3 km (103x103 grid points), and 444 m (106x106 grid points) respectively.

D03 covers the whole isthmus area, with Chivela pass approximately in its center, while the highest resolution domains d04 and d05 are slightly displaced to the south and east. D04 includes Chivela pass and the section of the Sierra Madre de Chiapas range east of it, with heights reaching about 2000m. For its part, the finest grid D05 encompasses the southernmost hills to the east of Chivela pass and the coastal plain at their base. This domain configuration focuses the area of interest of the study on the Pacific side of the Tehuano wind path, including the exit of the mountain gap and the gradually rising mountains east of it,

around the only two available observational sites for validation, labeled as MET1 and MET2 in Figure 1b.

All the domains have 70 hybrid-sigma vertical levels, 8 of which lie within the first 200 m above ground, at about 16, 46, 71, 96, 122, 147, 173 and 198 m height. The hybrid sigma-pressure vertical coordinate follows the terrain near the surface and gradually transitions to constant pressure at higher levels. The benefit of this vertical coordinate system is a numerical noise reduction in the upper-layers over mountains (Powers et al., 2017). We maintain this fine vertical grid spacing in all the

domains, to capture as wide a range of motions as possible over the depth of the boundary layer.

Land use information for d04 and d05 is obtained from the ESA CCI (European Space Agency Climate Change Initiative) database (Bontemps et al., 2013) with a resolution of 300 m. The terrain elevation used comes from the ASTER Global Digital Elevation Map (GDEM) from USGS (United States Geological Survey)(Slater et al., 2009) with a resolution of 30 m. In the





other domains, terrain and land use data are from the WRF global standard database, both at 30" resolution for d03 and 2' for d02 and d01.

We simulate a 36-hour period, from 2013-12-23 12:00 to 2013-12-25 00:00 UTC, which registers DSWS conditions in the observational data. Regarding the main physics options, the simulations use the tropical suite configuration (Table1), introduced
in WRF version 3.9., except for the planetary boundary layer, which is parametrized by the Shin-Hong scale-aware scheme (S-H) (Shin and Hong, 2015). The next table summarizes this physics configuration used.

| Microphysics | Hong and Lim (Hong and Lim, 2006) |
|---|---|
| Cumulus | Zhang and Wang. (Zhang et al., 2011) *disabled in d04 and d05 |
| Long wave radiation | RRTMG (Iacono et al., 2008) |
| Short wave radiation | RRTMG (Iacono et al., 2008) |
| Planet boundary layer | Shin and Hong (Shin and Hong, 2015) |
| Surface layer option | Revised MM5 surface layer (Jiménez et al., 2012) |
| Land-surface physics | Noah land-surface (Tewari et al., 2004) |

**Table 1.** Main physics parameterizations used.

The S-H planetary boundary layer option is more suitable for the high resolution of the innermost domain (444 m) because it helps to mitigate a double counting effect of the small-scale processes in gray-zone resolutions. Apart from this, this scheme provides a turbulent kinetic energy (TKE) diagnostic variable useful for our analyses.

## 2.2 Global model and real data

Global Forecast System (GFS) analysis data from the National Centers for Environmental Prediction (NCEP) is used as initial and boundary conditions for the WRF model, with a 3-h update interval. The horizontal resolution of this dataset for all variables is 0.5 x 0.5 deg, with 32 vertical levels ranging from 1000 to 10 hPa. The observational data used in this work is provided by the Mexican National Laboratory of remote sensors (https://clima.inifap.gob.mx/lnmysr/Estaciones/MapaEstaciones), col-
lected every 15 minutes at two meteorological stations whose location is presented in Table 2 and marked in Figure 1b as MET1 and MET2. Wind speed, wind direction, and temperature at 3 m height from these points are used for validation.

Model data are extrapolated from their native sigma levels to the height of the meteorological station using equation (3), which relates wind speed with friction wind speed, which is also a diagnostic variable in the model.

$$ws_z = \frac{ws_*}{K} ln(\frac{z}{z_0}) \tag{3}$$

Where $ws_*$ in the friction velocity, $K$ is the von Kármán constant, $z$ is the height and $z_0$ is the rugosity.




| | name | latitude (°) | longitude (°) | elevation (m) | height (m) |
|---|---|---|---|---|---|
| MET1 | Santiago Niltepec | 16.5535 | -94.6439 | 65 | 3 |
| MET2 | Ixhuatan | 16.3673 | -94.4717 | 18 | 3 |

**Table 2.** Weather stations position

## 3 Results and discussion

### 3.1 Synoptic and Mesoscale situation

Figure 2 shows the synoptic situation, from GFS analysis data, in North and Central America 18 hours before (a) and at the end of the period of study (b).

<p style="text-align:center">2013-12-22 18:00 UTC           2013-12-25 00:00 UTC</p>

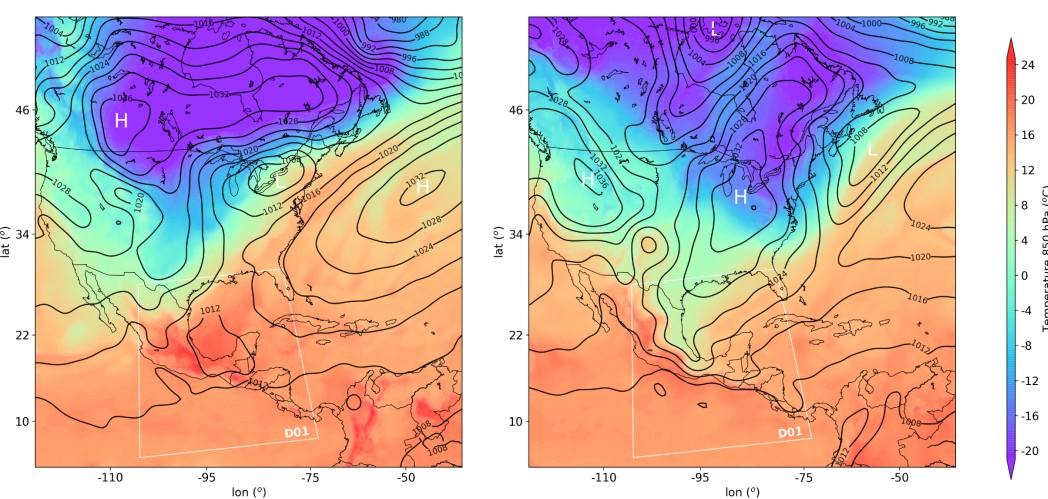

**Figure 2.** (a) 850 hPa temperature (°C) and sea level pressure (hPa) from GFS 0.5 Analysis data at 2013-12-22 18:00 UTC. (b) Same as (a) at 2013-12-25 00:00 UTC. D01 simulation domain is represented with a white square in both cases.

The large-scale setting is typical of Tehuantepecer wind episodes, where an Arctic air mass east of the Rockies pushes south across the Great Plains, with its leading edge reaching first the Gulf of Mexico (on December 22, Fig 2a) and then as far south as the bay of Campeche (one day later, Fig 2b), the result of cold air damming east of the Sierra Madre Oriental range in Mexico. The equatorward displacement of the associated high-pressure system on the wake of the cold front creates a strong

10 pressure gradient across the Isthmus of Tehuantepec, ultimately producing the strong mountain gap winds through the low elevation of Chivela pass. The general situation favoring Tehuantepecer winds persists for about 6 days; more pronounced earlier on.





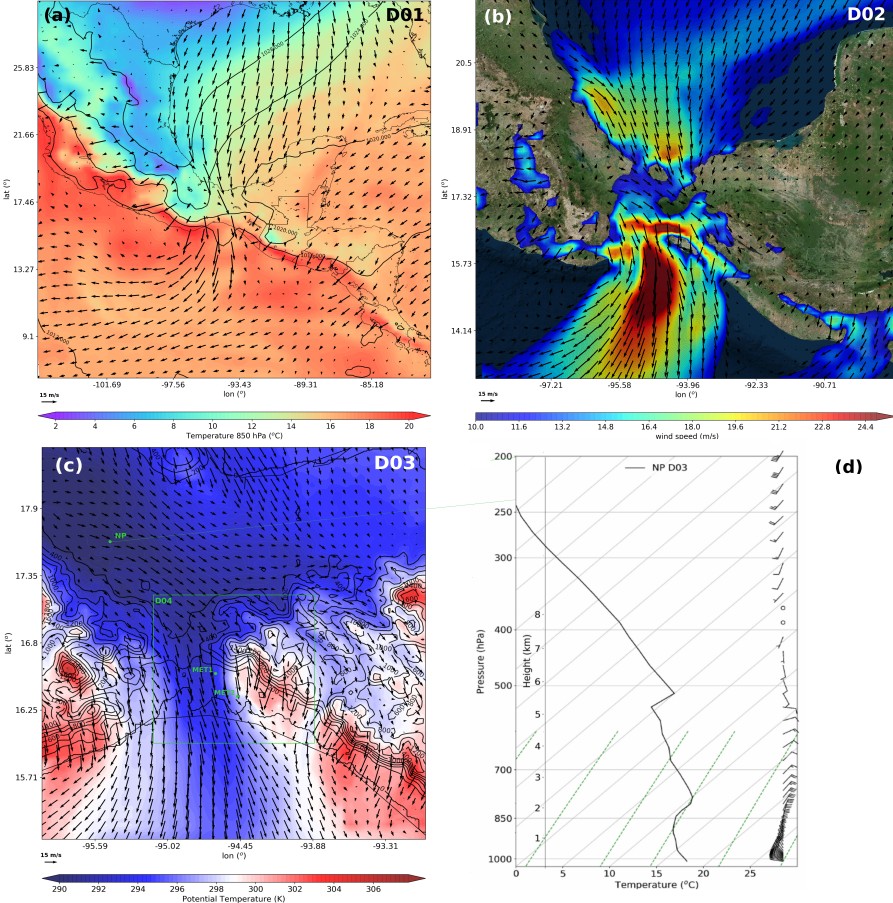

**Figure 3.** 2013-12-24 03:00 UTC (a) 850 hPa temperature (° C), sea level pressure (hPa) and wind arrows in the parent grid d01, (b) wind speed (values>10 m/s) and wind arrows at $\sigma$=3 (about 70m above ground) in d02, and (c) topography (m, contours) and potential temperature (K, shades) and wind arrows at $\sigma$=3 in d03.

Figure 3 shows the mesoscale conditions of the fully developed extreme wind episode at 03:00 UTC of 2013-12-24, from the WRF simulation. In the coarser grid D01 (36 km spacing, outlined in white in Figure 2) the cold air damming by the Sierra Madre Oriental mountains is clearly apparent, with the northerly cool air mass intrusion extending to the bay of Campeche, from where it is funneled across the Isthmus of Tehuantepec (Figure 3a). The higher resolution of the nested grids shows in greater detail the structure of the gap winds. Figure 3b, from the first nest D02 (12 km grid spacing), depicts the wind field (arrows) highlighting in shades the values above 10 m/s at about 70 m above the surface (sigma level 3), the approximate height of wind turbine hubs. The strong Tehuano outflow from Chivela pass reaches velocities of 25 m/s as it fans out for more than 100 km into the Pacific Ocean. High wind speeds are not only restricted to the outflow of the mountain gap itself; the simulation results suggest that they also occur in the mountains west and especially east of Chivela pass. The potential temperature field on the same sigma=3 level at the enhanced resolution (4km) of the next nested grid D03 (Figure 3c) illustrates how the stable cold

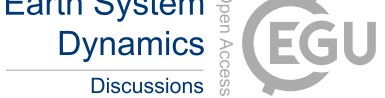



air mass to the north surmounts the lower hills neighboring the pass, particularly those to the east where elevation increases more gradually. Acceleration on the top of these mountains and to their lee is thus likely related to flow thinning and wave development, which can potentially result in strong downslope winds, rotors and hydraulic jumps. The model sounding at location NP on the Gulf of Mexico side of the Isthmus (Figure 3d), shows the depth of the cold air pool, defined by the

5 inversion existent at about 800hPa or 2500m in the temperature profile, indeed above the aforementioned mountain tops. The inversion associated to the subsidence within the high pressure system aloft is also clearly apparent just below 500hPa. Above this level, winds are weak and veer from being southeasterly to southwesterly in the upper troposphere. Below 500 hPa, winds back from an easterly to a northeasterly direction at about 800hPa, and more strongly in lower levels, becoming westerly at the surface, indicating intense cold air advection. There is a pronounced reverse wind shear in the lower troposphere.

As mentioned earlier, the only available observations in the area are from stations MET1 and MET2, whose position is marked in Fig.3c. They are both located on the Pacific coastal plain, south of the mountains bordering Chivela pass to the east; MET1 closer to the relief and further west than MET2. The wind speed time series covering the entire Tehuano episode for both stations is shown in Figure 4. Wind speeds are low in the previous days and show a daily cycle, likely linked to sea breezes, more clearly evident in station MET2 closer to the coast. The situation changes after about December 23 at 6

UTC, when picking up intensity, wind speeds become more constant throughout the day, the signature of a Tehuano wind occurrence. At about 0 UTC on the $29^{th}$, the episode decays and winds go back to local breeze regimes. The extent of the simulated period covers the first 36h of the event (shaded in Fig.4), corresponding with its highest intensity in both locations. These observations away from Chivela pass show evidence, as the simulations suggest, that the neighboring mountains may also induce strong winds, even though certainly not as far reaching as the mountain gap does.

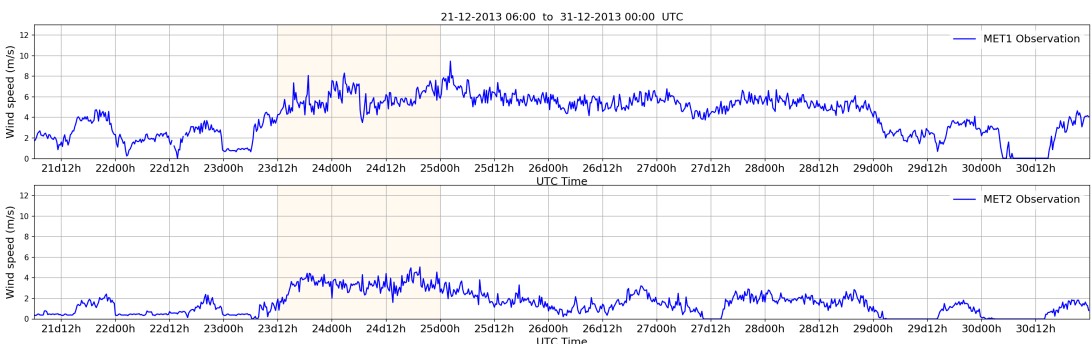

**Figure 4.** Wind speed (m/s) time series for observation stations MET1 and MET2 (locations marked in Figure 3). The orange shaded area corresponds to the period simulated.



## 3.2 Upstream-Downstream structure

Figure 3 in the previous section shows the general structure of the Tehuano wind event, produced by the cold air intrusion from the north and orographic forcings at different scales. In this section, we focus on the fine scale structure of the flow acceleration across the Isthmus, and more precisely, on that occurring on the westernmost section of the Sierra Madre de Chiapas mountains

bordering Chivela Pass, apart from the well-known strong gap wind jet. This is the area covered by the d04 domain of 1.3 km resolution (highlighted in green in Fig. 3c), encompassing with 137 km from north to south the wind flow path before and after crossing the mountains. Figure 5 depicts from d04, latitudinal cross-sections (red lines in Figure 1b) facing east (south to the left, north to the right) of different variables at the longitude of the two validation points MET1 (Fig. 5a-f) and MET2 (Fig. 5g-l).

The tight isentropes on the windward side of the mountains (to the right in Fig. 5a, d, g and j), indicate a quite similar stable stratification in the whole lower tropospheric column in both cases; stronger in the layers below about 2.5 km and weaker above. The temperature profile in Fig. 3d evidences that the latter level corresponds with the depth of the cold air. Winds are northerly in these lower layers to the north of the mountains, with somewhat higher speeds above 20m/s upstream from MET1 than further east, north of MET2. In both cases, winds above the cool pool are much weaker, and back to an easterly

component at about 4000m and above. The latter height is therefore a critical level, since the cross-mountain flow component becomes null; thus, triggered gravity waves will break and dissipate when they approach it. Markowski and Richardson (2010) outline seven conditions conducive to DSWS, albeit not all of them absolutely necessary. These conditions are: a mountain with steeper lee slope (1) crossed by strong winds (> 15m/s) (2) mostly normal to the barrier (3). A stable layer above the top and less stable above that (4) with cold air advection and large-scale subsidence to maintain the stability (5). Apart from this,

reverse wind shear above (6) and no cool pool in the lee (7), is also desirable. These conditions are all perfectly met for both locations analyzed, as discussed previously, and indeed intense downslope windstorms occur in both cases.

The stably stratified cross barrier flow displays wave activity from early on, and wave breaking, as the vertical isentropes suggest (Fig. 5a and d), enhances turbulent mixing (Fig. 5c and f) and yields a region of weak stability and reverse flow immediately downwind from the mountain crests (Fig 5g and j). In both cases, isentropes on the windward side sink sharply

under these layers of low stability on the lee side, much more pronouncedly for the tallest mountain (Fig. 5j). Encompassing the well mixed region to the lee, a split streamline develops (Smith, 1985b), and below its lower branch there is flow thinning and a significant increase in wind speed. The particular features existent on the lee side differ, however, depending of the height of the topographic obstacle. The strong accelerated flow bounded by intense turbulence extends for many kilometers downwind from the lowest mountain (Fig 5i), while ending with a hydraulic jump and rotors at the foot when the barrier is higher (Fig

5j). The formation of either of these lee wave events is related to the Froude number upstream (Smith, 1989).

Figure 6a plots the Froude number (Equation 1), calculated using the average of the variables at the first 5 sigma levels, approximately between 16 and 120m above ground, at the crest, H=936 m for MET1 case and H= 1736 m for MET2. For the lowest mountain, $Fr \approx 2.5$ during nighttime and even higher at some other times during the day, indicating supercritical conditions. The Brunt-Väisälä frequency is between 0.020 and 0.025 $s^{-1}$. The generated mountain waves have relatively short



**Figure 5.** d04 vertical cross sections (a,d,g,j) potential temperature (K, contours), wind speed (m/s shades), and wind barbs referenced to the orientation of the cross section plane, (b,e,h,k) vertical wind component W (m/s) and (c,f,i,l) wind speed (m/s) isolines and turbulent kinetic energy (TKE, J/kg, shades). First (a,b,c) row for MET1 and second row (d,e,f) for MET2 correspond to the initial stages of the episode on 2013-12-23 15:00 UTC. Third row (f,h,i) at MET1 and fourth (j,k,l) row at MET2 are for the fully developed events on 2013-12-24 03:00 UTC.

wavelength and small amplitude and a modest hydraulic jump that propagates downstream can be seen at the initial stages of the episode at 15 UTC December 23 (Fig. 5a). 12 hours later, during nighttime, the aforementioned strong jet extending for tens of kilometers downwind is fully formed, with values above 35 m/s at about 750m above ground and strong turbulence at the surface and in the layers above the jet, where stability is much reduced. The temperature profile upwind (at 3 UTC December 24 and location UP1 in Fig 5) shows stable conditions for the lowest troposphere, specially marked at crest level,





below the inversion at about 2500 m signaling the depth of the cool pool. At the observation location MET1, downwind from the mountain, the lowest layers up to about 900 m are well mixed (slightly lower, up to 750m further downstream), the result of the intense surface turbulence. Above the latter height and up to about 1500 m elevation, a strongly stable layer exists corresponding with the aforementioned packing of the isentropes (and streamlines). This is where the highest wind speeds are

found. Stability is much reduced further high, in the region encompassed by the dividing streamline, and winds are rather weak and have an easterly component for the most part, parallel to the mountains.

For the higher mountain north of MET2, $Fr \approx 1$ consistently in the period, indicating a critical flow regime, prone to the formation of HJs (Vosper2006 [5B]). The Brunt-Väisälä frequency is between 0.012 and 0.015 $s^{-1}$ and the generated waves have higher amplitudes than in the MET1 case. Wave overturning and breaking is also much more pronounced (Fig 5d) and

the forming well mixed region to the lee of the crest is deeper. Isentropes and streamlines that sink underneath this region are packed in a very shallow layer on the lee slope of the mountain, generating an intense downslope windstorm with speeds above 35 m/s at the surface. These strong winds end abruptly at the foot of the hill, where the flow transitions to subcritical conditions and a marked stationary hydraulic jump forms, with vertical wind speeds of 6 m/s. A rotor extending from the jump to the location of observation station MET2 is also evident in Fig. 5l. The temperature profile upstream is very similar to that of the

MET1 case, but downwind from the mountain at location MET2, lacks the strongly stratified layer present in the case of the lower mountain.

Results from the innermost nested grid d05 with the finest resolution (Fig 7) suggest that trapped lee waves develop in the MET1 case within the high stability layer where the strongest winds are found, just below the low stability region and the critical level aloft that prevent their vertical propagation. This wavelike pattern is a common feature in DSWS periods (Pokharel

et al., 2017b; Hertenstein and Kuettner, 2005) and fully formed 12 h later (Fig. 7c), extends for more than 100 km downstream aligned with the general orientation in the northwest-southeast direction of the Sierra in the region. Waves are absent further east in the MET2 cross section, where the topographic barrier is higher and a stationary HJ forms instead, as discussed above.

Our results suggest that during Tehuano wind events, the Pacific side of the Isthmus of Tehuantepec east of Chivela pass is very prone to host extreme winds' phenomena. The formation of DSWSs in the area increases the impacts of the already strong

mountain gap winds.

### 3.3 Validation

Finally, we contrast our simulation results with the very few data available for validation at meteorological stations MET1 and MET2. The two plots in Figure 8 compare the simulated wind speed (in d04 and d05) with observations from both stations.

Wind speed results from d04 and d05 at location MET1 are quite similar, and fare well with respect to observations (Figure

8a), slightly better so those from d05, with a mean absolute error (MAE) of 1.55 m/s (Table2). The similarity in the low mean error between both simulations and their high correlation throughout the period are due to the nature of the event in that area, an intense and mostly steady jet that the d04 domain resolution (1.3 km) is already capable of resolving accurately. However, results in MET2, which registers the HJ situations, present more differences between d04 and d05 (Figure 8b) and there is a significant improvement in d05 with respect to its parent domain d04. Wind speeds in d04 are overestimated (Mean error ME =



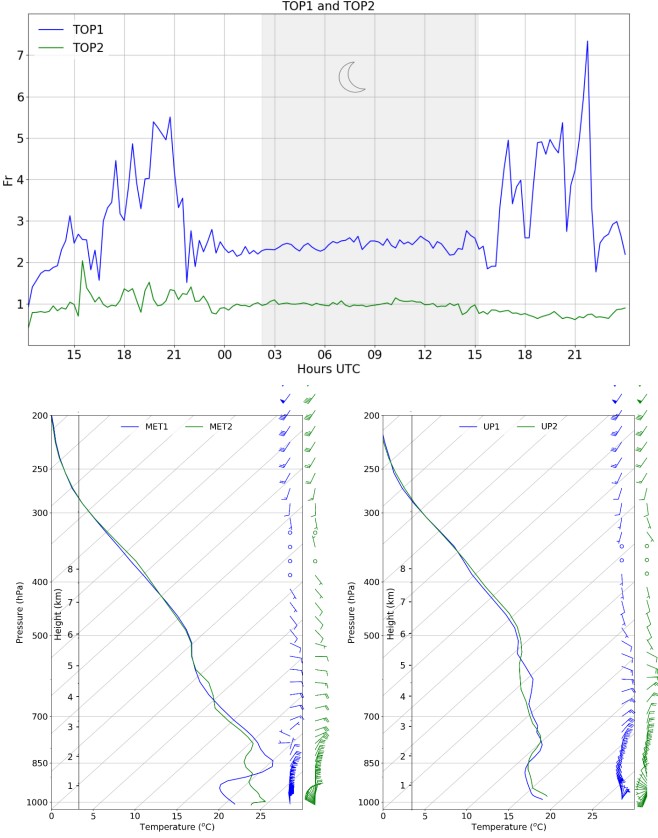

**Figure 6.** ((a) Froude number at the top of the mountain, north of MET1 (TOP1) and of MET2 (TOP2). The grey zone represents night time. (b) Vertical temperature profiles at MET1 and MET2 and (c) at their corresponding upstream points UP1 and UP2.

2.76 m/s), and present a daily cycle that is absent or very subtle in the observations. The complexity and fast variability of HJs formation in this area are better resolved in the higher resolution grid, which perhaps reproduces more accurately the stagnant flow and rotor formations downstream from the HJ. With regard to wind direction, errors are small for MET1 and significantly higher for MET2, due to the same reasons. As in the case of wind speeds, wind direction results from the finer grid d05 are

5    also better than in d04. Temperature errors are equal or below 1 K in both locations and domains, hence the surface thermal evolution is well captured.

   Lee waves can promote orographic cloud formation at different scales, depending on the amplitude of the wave and elevation (Armi and Mayr, 2011; Szmyd, 2016). Model results in d05 suggest that lenticular clouds form at the crests of the trapped lee waves depicted in Figure 7. Figure 9a shows a 3D representation of the modeled cloud water mixing ratio at 2013-12-24 15:30

10   UTC in d05. Cross sections of wind speed at the surface observation locations MET1 and MET2 as in Fig 5 and Fig 7 are also included for reference. A 2D view of the same cloud mixing ratio variable and wind arrows at sigma level 17 (about 1400 m





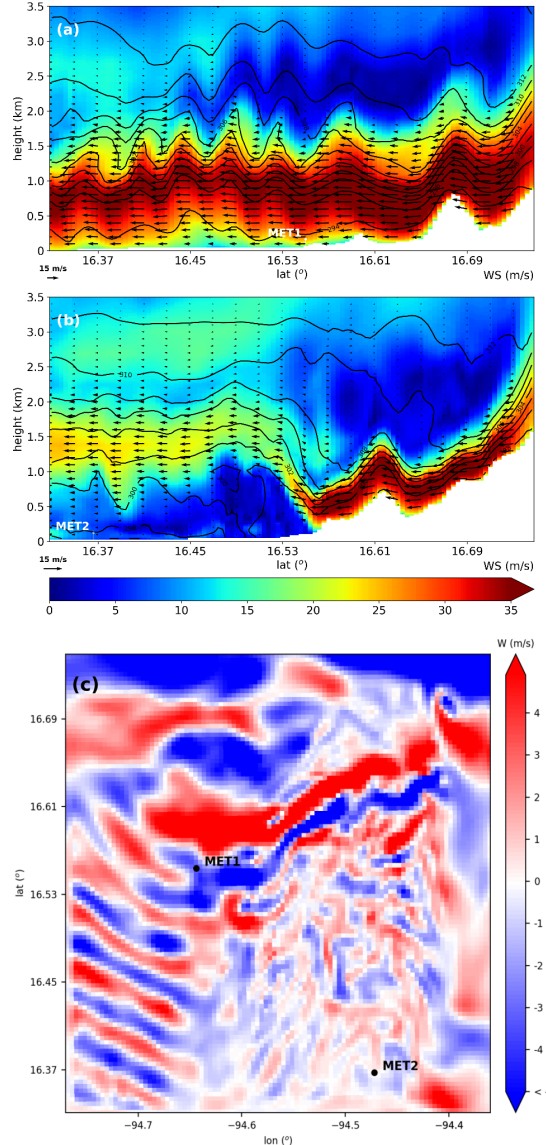

**Figure 7.** As in Fig. 5g and 5j but for d05, vertical cross sections of potential temperature (K, contours), wind speed (m/s shades), and wind barbs referenced to the orientation of the cross section plane at 2013-12-24 03:00 UTC for (a) MET1 and (b) MET2. Vertical wind speed (m/s) in d05 at sigma level 17 (about 1400m above ground) when the trapped lee wave pattern in the region is fully formed at 15:30 UTC 2013-12-24.

above ground), revealing existing clouds, are depicted overlaying a satellite image of the area. An actual satellite image from Geostationary Operational Environmental Satellite - R Series (GOES-R) (http://www.goes-r.gov/education/docs/fs_imagery. pdf) around the same time is shown for comparison, indicating that remarkably similar mountain wave cloud formations



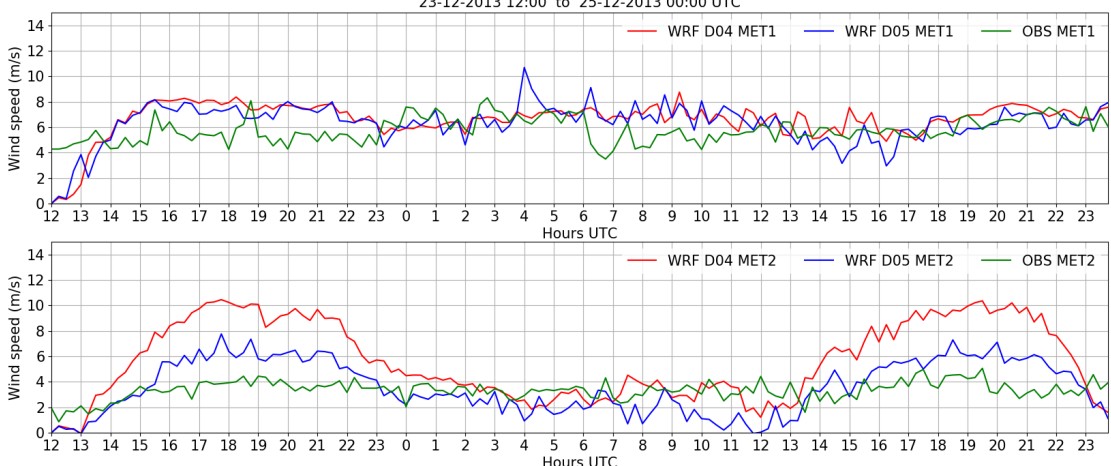

**Figure 8.** (a) Wind speed comparison among observations at MET1 (green), and model output from d04 (red) and, d05 (blue). (b) Same for station location MET2.

|  | WS MAE (m/s) | WS ME (m/s) | WD MAE (°) | T MAE (K) | T ME (K) |
|---|---|---|---|---|---|
| **MET1 D04** | 1.82 | 1.26 | 16.15 | 0.85 | -0.80 |
| **MET1 D05** | 1.55 | 0.80 | 13.63 | 0.77 | -0.71 |
| **MET2 D04** | 2.76 | 2.35 | 27.87 | 0.71 | 0.01 |
| **MET2 D05** | 1.31 | 0.18 | 24.07 | 1.02 | 0.61 |

**Table 3.** Wind speed (WS), wind direction (WD), and temperature (T) mean errors (ME) and mean absolute errors (MAE) at MET1 and MET2 locations during the simulated period and for the two higher resolution domains (d04 and d05).

were indeed observed in the area. The actual existence of these lenticular clouds with the same location and pattern as in the simulation further validates the model results.

## 4 Conclusions

In the present work, we studied lee wave phenomena occurring during Tehuano events on the Pacific side of the Isthmus of Tehuantepec using WRF high-resolution simulations. Orographic forcings at different scales result in the well-known gap wind jet off Chivela pass, but also in downslope windstorms and hydraulic jumps in the neighboring mountains. We analyzed these phenomena in an episode in December 2013 having the typical genesis of Tehuantepecer wind events. An Arctic air mass in North America pushed as far south as the bay of Campeche due to cold air damming east of the Rockies continuing to the east of the Sierra Madre Oriental range in Mexico. The displacement of the associated high-pressure system on the wake of the cold front created large pressure differences across the Isthmus of Tehuantepec, ultimately producing the strong mountain gap





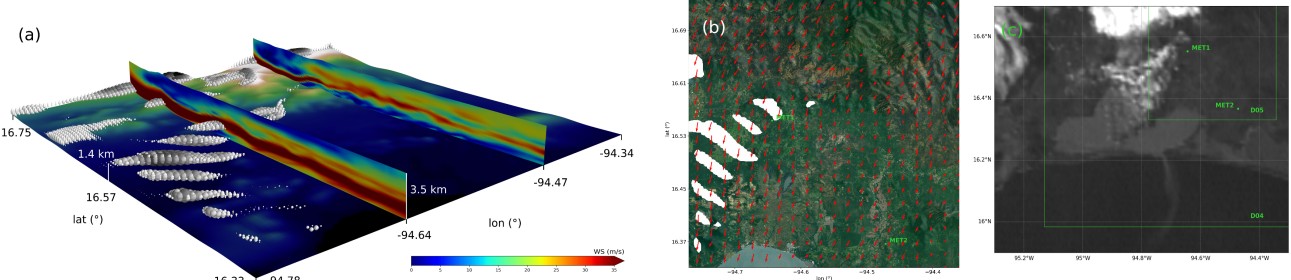

**Figure 9.** (a) Cloud water mixing ratio 3D representation in d05 at 2013-12-24 15:30 UTC. The two cross sections show the N-S wind profile at the longitudes of the meteorological stations MET1 and MET2. (b) Satellite image of the terrain in d05 and cloud water mixing ratio (white shades) and wind arrows at about 1.4 km above ground.(c) GOES-R satellite image on 2013-12-24 20:30 UTC, revealing very similar lenticular cloud formations in the same locations.

winds through the low elevation of Chivela pass. The model simulates these intense winds, blowing with speeds at the surface of more than 25 m/s that extend for many kilometers from the mountain gap, fanning out well into the Gulf of Tehuantepec, as it is commonly observed in Tehuano events (Steenburgh et al., 1998).

The depth of the cold surge on the coastal plains of the Gulf of Mexico side of the Isthmus is about 2500m, therefore thick

enough to surmount the lower elevations to the west, and especially to the east of Chivela pass. The flow over these mountains results in intense downslope wind storms to their lee, with the generation of intense turbulence, hydraulic jumps and rotors, depending on the particular height of the topography. We focus on two locations of different barrier elevation from where there are surface observations downwind: one of 963m closer to Chivela pass and another further east, with height increasing to 1736m

The thermodynamic characteristics of the air mass are rather uniform upwind of both mountains, with strong stability within the cool pool and weaker above, and intense northerly winds that back and weaken aloft to a more easterly component parallel to the barrier. The critical level where the cross-mountain wind component becomes zero, inhibiting wave propagation, is about 4000m. Mountain waves are generated in both cases, with smaller amplitudes for the lower mountain, where the Brunt-Väisälä frequency at crest height is between 0.020 and 0.025 $s^{-1}$, than for the higher mountain, where the Brunt-Väisälä frequency is

about half. Wave breaking produces mixing and generates a region of low stability to the lee of the mountains, which is deeper where the waves have higher amplitude. The Froude number is around 2.5 at crest height in the lower barrier and the flow presents a supercritical behavior. The region of low stability to the lee of the mountain lies above about 1500m and leads to a packing of the isentropes and streamlines underneath, resulting in strong stability and flow acceleration. An intense jet develops with wind speeds of 35 m/s at about 750m above ground extending for tens of kilometers downwind from the mountains. Wind

speeds are reduced closer to the surface due to intense turbulence. Trapped lee waves form at about 1500m, just below the well mixed layer aloft that prevents their vertical propagation. The Froude number decreases to about 1 further east as elevation rises and the flow presents a critical regime. Isentropes on the windward side of the mountain sink much more pronouncedly under the wider mixed layer generated by wave breaking to the lee, and are tightly packed in a shallow layer above the surface.

This generates an intense wind storm on the lee slope of the mountain, with surface wind speeds up to 35 m/s. The accelerated flow down the mountain ends abruptly at its foot, where the flow turns to subcritical state and a marked stationary hydraulic jump forms, with vertical velocities of 6 m/s. A rotor circulation develops further downstream from the jump.

Only limited observations are available to validate our model results. Errors in surface wind speeds, directions and temperature are small at the only two stations available on the Pacific coastal plain downwind from the mountains. In addition, lenticular clouds similar in location and pattern to those produced by the model, are apparent in satellite imagery of the day of the event, providing valuable indication that the mountain lee wave phenomena simulated indeed correspond to a real scenario.

Our model results suggest that when the cold air mass intruding from the north on the Gulf of Mexico side of the Isthmus is thick enough to surmount the mountains, extreme wind events develop in the area during Tehuano events beyond the gap wind jet. These include downslope wind storms and hydraulic jumps, which are intense and highly turbulent flows that can have a substantial impact on the existent wind farm industry in the region. It is likely that the depth of the cold pool and how it compares with topographic barrier height, is a key factor determining the extent, location and intensity of these lee wave phenomena and whether they take place at all.

*Author contributions.* TEXT

*Competing interests.* TEXT

*Disclaimer.* TEXT

*Acknowledgements.* We would like to thank Mexican National Laboratory of remote sensors (INAFAP) (http://clima.inifap.gob.mx/lnmysr/) for the valuable real data provided, necessary to validate all the study. The model forecast simulations and development of the data analysis was performed at the Centro de Supercomputacion de Galicia (CESGA) (http://www.cesga.es/). Their computer facilities and support have been indispensable to carry out this project. Finally, we would like to acknowledge the Non-Lineal Physics Group of Universidade de Santiago de Compostela (http://www.usc.es/en/investigacion/grupos/gfnl), which is where this whole project has been developed.



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
