# Peer review of "Downslope windstorm study in the Isthmus of Tehuantepec using WRF high-resolution simulations"

_Earth System Dynamics, 2019_

## Referee Comment (RC1) · Anonymous Referee #1 · 18 Mar 2019

This manuscript is a product of hard working and written in detail. However, if it is written concisely it will be much better. I think if it is revised a bit with editing and by incorporating some provided information it will be considered for publication.

Page 1 Line 26: Re-write this sentence concisely: "Wave breaking creates a well-mixed region to the lee of the obstacle that induces flow separation; the generation of a dividing streamline between undisturbed flow above and trapped energy and flow analogous to hydraulics in the lower surficial branch (Smith, 1985b)."

Page 3-4 Section 1.1 is a big with repeating same expression time to time. As this is a scientific paper shorten being on a specific on Mountain wave phenomena, hydraulic

analog and relation with the Froude number.

Page 4 Line 20: "Fr in (Eq3)" is it equation 3? If so, there is no relation of Fr. Correct it.

Page 6 Line 20; Replace "rugosity" by surface roughness (m)

Page 7 Line 6: "wind episodes". Where is wind? Figure 2a and 2b say they are 850 hPa level temperature and sea level pressure and also there are no any wind arrows. So, include wind.

Page 9 Line 4: "Isthmus (Figure 3d)". There is no caption of Figure 3d. Add this including its exact position (lat./long) so that it will can be compared with aforesaid matters.

Page 9 Line 5: "inversion existent at about 800hPa or 2500m". Seems from about 2000 to 2500 m. correct this. Looks sounding has multiple PBL and discontinuous stratification which refers an importance of study of high resolution data sets. So, mention this kind of information to highlight your approach of methodology.

Page 10 Line 10: "The temperature profile in Fig. 3d evidences that the latter level. . . ".. What does latter level mean as it is hard to understand? If you are referring a presence of stable layer below 2.5km Figure 3d shows there is a presence of static instability, not stability below 2 km. So, regarding this issue rewrite this.

Page 10 Line 31: "Figure 6a plots the Froude number (Equation 1)" How did you find "$\theta 0$" (method) to calculate N for Froude number?

Please mention figure nos. in following sentences as it will be easy to follow these criteria:

Page 10 Line 17-21: These conditions are: a mountain with steeper lee slope (1) (Figure no. ??) crossed by strong winds (> 15m/s) (2) (Figure no. ??) mostly normal to the barrier (3) (Figure no. ??). A stable layer above the top and less stable above that (4) (Figure no. ??) with cold air advection and large-scale subsidence to maintain the

stability (5) (Figure no. ??) . Apart from this, reverse wind shear above (6) (Figure no. ??) and no cool pool in the lee (7) (Figure no. ??), is also desirable. These conditions are all perfectly met for both locations analyzed, as discussed previously, and indeed intense downslope windstorms occur in both cases.

Page 10 Line 31: Where is Figure 6a as there is no label a, b, c of Figure 6. So, it is hard to follow.

Page 14: Insert figure no. "c" in Figure 7.

How about the subcritical flow before the supercritical flow? Did you observe it within your region of interest before the supercritical flow? If so mention them with a value of Froude number so that it could be better to relate how the subcritical flow was converted to supercritical flow as downslope wind.

---

## Referee Comment (RC2) · Anonymous Referee #2 · 19 Mar 2019

The paper discusses simulations of one downslope wind event with the limited-area model WRF using multiple nests down to very high resolution (444 m). The study is mostly well written and interesting but needs some substantial editing. I also think it would gain a lot if an additional small sample of Isthmus events could be simulated and discussed (perhaps not using the innermost expensive WRF nest). Doing so would allow to inter-compare events and discuss similarities/differences. Still, the paper might be publishable without doing so.

Additional minor comments:

Page 4, line 21: missing equation P5, l2: Figure 2c missing P6, l 13: 0.5 deg x 0.5

deg P6, l20: rugosity is surface roughness length Table 2: station elevations correct? Figure 2: Why before and after the event and not at peak intensity? P9, l3: In the results section there should be no hypothesis ("which can potentially) P9, l10: whose positions P10, l20: Where can this perfect match be seen? Figure 5: It is difficult to understand what is what cross section? Mention "up1" in the caption. Figure 9b,c: Easier to compare if you use the same domains? Did you discuss the missing northern stratiform cloud deck in the simulations?

---

## Referee Comment (RC3) · Anonymous Referee #3 · 19 Mar 2019

General: This paper reports on simulations of terrain induced wind systems in Mexico, and is novel in the respect that I have not seen much work in this particular region. The question that I have been wrestling with is however: "Is this novel science?"

The simulations are carried out for one single events, and while the authors may feels this was a typical event, the reader is not given much information other than some verbal arguments. So this remains a case study, and does not even claim to be general. The type of flows that are described (mainly downs-slope wind-storms and hydraulic jumps) occur at many places around the world and has been studies extensively both from observations and with models. It is well known that numerical models perform

well in these type of "hydraulic flows"; hence it does not reveal anything new about the flow features other than such that are directly related to specific local features. The authors do not even try to show that this type of flows have special characters because of the geographic location, which closer to the equator than in most other studies. '

Hence, there is nothing new here and from that respect I should have recommended "reject". However, there is nothing fundamentally wrong with the study as such, so maybe that would be unfair. So I will leave to the editor to determine if the world needs one more paper about these events. I would tend to say "no", but in the case the editor say "yes" the paper still needs major revision.

I have tw0 main concerns: 1)he analysis if the model results is rather superficial and make many claims that are hard to substantiate from the graphic material presented. I'm sure this could be turned into a nice paper if the authors try to think a bit out of the box and perform an actual analysis of the model results, rather than just produce a few standard plots direct from the model data. Much of the theoretical background includes the planetary rotation; this is not commented on at all and here may be an opportunity to take a novel angle, comparing these results to higher-latitude cases. Just in general I would like to see a more in-depth analysis.

2) The main results are all together expected and reveals nothing about this flow that couldn't have guessed without the model simulations. Sure there are details here that no one could guess, but much of that detail is not harvested.

Finally, the language would benefit from editing by a native english speaker.

Specific: Make sure you define abbreviations the first time you use it, and stick withe abbreviation afterwards. HJ is not explained, and is used interchanging with "hydraulic jump".

Drop the entire section 1.1; this is textbook stuff and just take up space.

If you feel a need to validate the model, this should come before the experimental

results, not after. Moreover, the cloud evaluation is superficial to the point of being useless. Either drop it or develop it.

P3, l25: What do you mean by microscale?

Figure 1: Why are the two most high-resolution domain off center wrt to the gap?

P6, l3: How much of a spinup is required before the model physics is realistic?

P6, l18: This is incorrect; U* is not a wind speed; its a scaling parameter that depends on the vertical turbulent momentum flux. Also explain how the logarithmic wind-law is applied; with this formula, the wind just increases with height, so you need an anchor point.

Figure 2: What is the height of the 850 hPa wrt mountain crest? Why pressure levels at all, and not model levels? Note the warmest temperatures hanging on the southward facing lee slope; DSWS already happen here? Maybe the D01 domain is a bit small, or could have been located farther north, since there's a lot of uninteresting ocean south of the coast.

Figure 3: Results are impressive but not unexpected. Moreover, legend for panel (d) is missing.

P8, l8: What is the Rossby radius of deformation here.

P8, L9: Are you here referring to the cross-over in the T-shaped structure? Might this is the DSWS? There seems to be a tremendous along-flow convergence/divergence here.

P9, l2: Here you argue that this is a "flow thinning" event and not a gravity-wave break-ing downs-slope flow event, but later tit is the opposite.

P9, l4: And where is NP? Moreover, this analysis (d-panel) would have been much more informative if you had analysed the depth of the cold air and plotted it as a contour plot, showing its geographical distribution

P9, l5&6: If you want to use this type of sounding plot, you need to tell the reader what's on the axes. If the gray transverse lines are isotherms, there is hardly any inversions at all; to me the lwoer one looks more like an isothermal layer while the upper (subsidence) may be a weak one.

P9, l19: Awkward; what do you mean by "far reaching"?

Figure 4: Show the modeled wind speeds already here and save a plot later.

Section 3.2: drop first sentence; we just read about that, no need to repeat.

P10, l67; Awkward English; what is a "wind path"?

P10, line 19; "steeper" than what?

P10, l23-24 and elsewhere: I don't dispute the wave-breaking argument, but how can you see this here? There are no temperature-gradient reversals that I can see, nor is there any TKE aloft that would result from it. I would have expected to see at least truly vertical isotherms and elevated layers of TKE, or gradient reversals and no TKE.

P10, l28: What do you mean by "bounded by turbulence"?

P10, l31 The use of Fr is a powerful but yet blunt instrument to analyze these flows. I have two concerns here: 1) As the air has propagated up til crest of the terrain, Fr is already modified. The classical analysis by Durran, cited earlier, also uses the upstream Fr before the flow has hit the terrain; not that at the top of the hill. Hence I would have liked to see the truly upstream Fr instead; not the value that has already been modified. 2) Is it certain that the air reaching the observation stations actually comes from the point directly north of the station? A trajectory analysis would clarify the 3D dynamics of the flow.

Figure 5: To much information in too too many too small panels. In fact, you could easily get rid of one third, by plotting the TKE and w in the same panels, as you do with temperature and winds.

P11, l1: The position of the hydraulic jump, which is far from very distinct to begin with, hardly makes any propagation clear. Instead, analyze the position of the jump at different times and plot the position, and maybe also its strength, as a function of time. Maybe also as a function of Fr. From this the reader cannot really see any propagation.

P12, line 1 Use "indicating" instead of "signalling".

P12, l5; "further high" is awkward.

P12, l9: Can't see any wave overturning in these plots. This may be because mixing erodes overturning isotherms before they can be seen in the model output. In that case there should be TKE there, which I can't see either.

P12, l30: Awkward; "slightly ... d05".

P12, l34: Also awkward; "which situations".

Conclusions contain far reaching statements that cannot be substantiated by one cas study.

―――――――――――――――

---

## Author Comment (AC1) · 23 May 2019

*Supplementary material uploaded: PDF with answers in an easy-readable format.

This manuscript is a product of hard working and written in detail. However, if it is written concisely it will be much better. I think if it is revised a bit with editing and by incorporating some provided information it will be considered for publication.

We thank the reviewer for accepting the correction of this article; we appreciate his/her favorable opinion about the work. We agree with most of the comments and we provide next a point by point answer to the concerns presented.

[Figure]

1- Page 1 Line 26: Re-write this sentence concisely: "Wave breaking creates a wellmixed region to the lee of the obstacle that induces flow separation; the generation of a dividing streamline between undisturbed flow above and trapped energy and flow analogous to hydraulics in the lower surficial branch (Smith, 1985b)."

We will now rewrite this sentence more concisely, as follows:

"Wave breaking creates a well-mixed layer to the lee of the obstacle, which generates a dividing streamline separating undisturbed flow aloft and trapped energy and flow analogous to hydraulics in the lower surficial branch (Smith, 1985b)"

2- Page 3-4 Section 1.1 is a big with repeating same expression time to time. As this is a scientific paper shorten being on a specific on Mountain wave phenomena, hydraulic analog and relation with the Froude number.

We agree with the reviewer. In this section we aimed to put in context the physics behind the studied phenomena and introduce the hydraulic analog. However, it is true that it could be repetitive in some way and too extensive. In the revised version we reduce this section.

3- Page 4 Line 20: "Fr in (Eq3)" is it equation 3? If so, there is no relation of Fr. Correct it. We apologize for the misunderstanding, we wanted to refer to equation 1. We correct this typo in the revised version.

4- Page 6 Line 20; Replace "rugosity" by surface roughness (m)

We will replace the word rugosity by surface roughness (m) in the revised version, as per the reviewer's request.

5- Page 7 Line 6: "wind episodes". Where is wind? Figure 2a and 2b say they are 850 hPa level temperature and sea level pressure and also there are no any wind arrows. So, include wind.

In these lines we are referring to the synoptic setting leading to the Tehuantepecer wind

event. As discussed in the text, the cold air mass moving south from north America is instrumental for the development of Tehuantepecers, which is why Figure 2 shows 850 hPa and sea level pressure in a very wide domain. Winds are better shown with much more detail, focusing on the Isthmus, in Figure 3.

6- Page 9 Line 4: "Isthmus (Figure 3d)". There is no caption of Figure 3d. Add this including its exact position (lat./long) so that it will can be compared with aforesaid matters.

We apologize for our mistake. In the revised version we include the caption for this figure where the lat and lon of the sounding are explicitly mentioned. There is a line from Figure 3d to Figure 3c pointing to the location (NP) where the sounding is performed, but it is barely noticeable. We will mark it more clearly in the revised version.

7- Page 9 Line 5: "inversion existent at about 800hPa or 2500m". Seems from about 2000 to 2500 m. correct this. Looks sounding has multiple PBL and discontinuous stratification which refers an importance of study of high resolution data sets. So, mention this kind of information to highlight your approach of methodology.

Another reviewer has noted that more than an inversion it is an isothermal layer, also highly stable, but not really an inversion. We will correct this issue and also include a comment about the need for high resolution for a study of this event. We thank the reviewer for this suggestion.

8- Page 10 Line 10: "The temperature profile in Fig. 3d evidences that the latter level. . . "... What does latter level mean as it is hard to understand? If you are referring a presence of stable layer below 2.5km Figure 3d shows there is a presence of static instability, not stability below 2 km. So, regarding this issue rewrite this.

We meant "the temperature profile in Fig. 3d evidences that the 2500m height level corresponds with the depth of the cold air". The profile is stable above this level, but much less so than in the layers below. We agree that it can be confusing as written in

the original text, we will rewrite this statement in the revised version. We will also add dry adiabats in the sounding in Fig 3 to help identifying stable layers.

9- Page 10 Line 31: "Figure 6a plots the Froude number (Equation 1)" How did you find "$\theta 0$" (method) to calculate N for Froude number? Please mention figure nos. in following sentences as it will be easy to follow these criteria:

We mostly follow [1] [2] and [3] below. As per some other reviewer's request, we will now show the Fr number some distance upstream the obstacle. We calculate Fr using the average Brunt-Väisällä frequency and average wind speed for the layers from the surface to the level of similar elevation as the mountain. We will explain this clearly in the revised manuscript.

[1] Pokharel, B., Geerts, B., Chu, X., and Bergmaier, P.: Profiling radar observations and numerical simulations of a downslopewind storm and rotor on the lee of the Medicine Bow mountains in Wyoming, Atmosphere, 8, https://doi.org/10.3390/atmos8020039, 2017b. [2] Pokharel, A. K., Kaplan, M. L., and Fiedler, S.: Subtropical Dust Storms and Downslope Wind Events, Journal of Geophysical Research: Atmospheres, 122, 10 191–10 205, https://doi.org/10.1002/2017JD026942, 2017a. [3] Grubiši′c, V., Serafin, S., Strauss, L., Haimov, S. J., French, J. R., and Oolman, L. D.: Wave-induced boundary-layer separation in the lee of the Medicine Bow Mountains. Part II: Numerical modeling, Journal of the Atmospheric Sciences, p. 150904104933002, https://doi.org/10.1175/JAS-D-14-0376.1, http://journals.ametsoc.org/doi/abs/10.1175/JAS-D-14-0381.1, 2015.

10- Page 10 Line 17-21: These conditions are: a mountain with steeper lee slope (1) (Figure no. ??) crossed by strong winds (> 15m/s) (2) (Figure no. ??) mostly normal to the barrier (3) (Figure no. ??). A stable layer above the top and less stable above that (4) (Figure no. ??) with cold air advection and large-scale subsidence to maintain the C2 ESDD Interactive comment Printer- friendly version Discussion paper stability (5) (Figure no. ??) . Apart from this, reverse wind shear above (6) (Figure no. ??) and

no cool pool in the lee (7) (Figure no. ??), is also desirable. These conditions are all perfectly met for both locations analyzed, as discussed previously, and indeed intense downslope windstorms occur in both cases.

These seven conditions were all mentioned at some point in the previous discussions, but we agree with the reviewer in that it will be very informative to summarize here the figures that evidence them as a reminder. We will modify the text accordingly in the revised version of the manuscript.

11- Page 10 Line 31: Where is Figure 6a as there is no label a, b, c of Figure 6. So, it is hard to follow.

We totally agree, excuse us for the mistake. This will be corrected in the revised version.

12- Page 14: Insert figure no. "c" in Figure 7. How about the subcritical flow before the supercritical flow? Did you observe it within your region of interest before the supercritical flow? If so mention them with a value of Froude number so that it could be better to relate how the subcritical flow was converted to supercritical flow as downslope wind.

We will introduce the subfigure letter (c) in the caption. Concerning the second issue, we will show the Fr number upstream the mountain in the revised manuscript. FrïĆż1 for the highest mountain and FrïĆż2 for the lowest one. In both cases the Fr values are in the range indicating transitional conditions from subcritical to supercritical behavior in the flow, as discussed in section 1.1 in the Introduction. It is difficult to define a Froude number with the same meaning as in shallow water; we nevertheless observe subcritical flow behavior, in the sense that the flow thins and speeds up as it goes upslope in the case of the lower mountain (Fig 5a and 5g), but not so clearly for the higher one.

Please also note the supplement to this comment:
https://www.earth-syst-dynam-discuss.net/esd-2019-3/esd-2019-3-AC1-

supplement.pdf

---

## Author Comment (AC2) · 23 May 2019

*Supplementary material uploaded: PDF with answers in an easy-readable format + figures.

The paper discusses simulations of one downslope wind event with the limited-area model WRF using multiple nests down to very high resolution (444 m). The study is mostly well written and interesting but needs some substantial editing. I also think it would gain a lot if an additional small sample of Isthmus events could be simulated and discussed (perhaps not using the innermost expensive WRF nest). Doing so would allow to inter-compare events and discuss similarities/differences. Still, the paper might

be publishable without doing so.

We would like to thank the reviewer for accepting to review this manuscript; we appreciate the positive view of the work. We believe that the modifications suggested will improve the manuscript.

We agree with the reviewer's suggestion in that details about additional Isthmus events could contribute to their better understanding. We have simulated other Tehuantepecer cases occurring in 2013 and we have seen downslope windstorms and hydraulic jumps developing in the mountains east of Chivela pass, as those shown in the paper. The one we selected is the most intense event of the year. We nevertheless prefer to just focus the study in one clear case, given that these phenomena have never been reported before in the area. We leave for future work a more careful study with a sufficient number of cases to investigate the variability these extreme events.

In the revised manuscript we will include in the conclusion section a comment about our preliminary results for other events and the need for a more comprehensive study with a sufficient number of cases to investigate the variability associated with these extreme phenomena.

Next, we address each of the minor concerns presented by the reviewer

Additional minor comments:

1- Page 4, line 21: missing equation

It is true, there is a mistake in this line. In this case we want to refer to Equation 1, which is the Froude number calculation. We will correct it in the revised version.

2- P5, l2: Figure 2c missing

Here we were referring to Figure 1b, where d04 and d05 domains are displayed. We correct this typo in the revised version.

3- P6, l 13: 0.5 deg x 0.5 deg

We agree, we will perform the suggested correction in the revised version. 4- P6, l20: rugosity is surface roughness length

The reviewer is right; we will replace "rugosity" by "surface roughness length" in the revised version of the manuscript.

5- Table 2: station elevations correct?

Yes, it is correct, the elevations are low because both stations are on the coastal plain at the foot of the mountains. The observational data were provided by the Mexican National Laboratory of remote sensors. (https://clima.inifap.gob.mx/lnmysr/Estaciones/MapaEstaciones)

6- Figure 2: Why before and after the event and not at peak intensity?

A map similar to those in Fig. 2 but at peak intensity and focused on the Isthmus is shown in the ensuing figure Fig 3a in the paper. But we agree with the reviewer in that in Fig 2 it makes more sense to show the larger scale situation also at peak intensity than later during the episode. We will now replace Fig2b by the map below, corresponding to 24 December at 03 UTC.

FIGURE 1 ATTACHED

7- P9, l3: In the results section there should be no hypothesis ("which can potentially)

We will correct the sentence (also including another reviewer's suggestion) as follows: "As shown in the following section 3.2, acceleration on the top of these mountains and to their lee is related to gravity wave activity, which results in the development of strong downslope winds, rotors and hydraulic jumps."

8- P9, l10: whose positions

The correction will be made as suggested.

9- P10, l20: Where can this perfect match be seen? The fulfillment of the listed conditions is supported by figures and discussed previously in the text. But we agree with the reviewer in that it should be reminded here. We will change the paragraph as follows:

"These conditions are: (1) an asymmetric mountain with steeper lee than windward side (Fig 5), (2) crossed by strong winds (> 15m/s) (Fig 5), (3) mostly normal to the barrier (Fig.3, 5). (4) A stable layer above the top and less stable above that (Fig 3d, 6c), (5) with cold air advection and large-scale subsidence to maintain the stability (Fig 3d, 6c). Apart from this, (6) reverse wind shear above (Fig 3d, 6c) and (7) no cool pool in the lee (Fig 3a, 6b), is also desirable. These conditions are all perfectly met for both locations analyzed, as discussed previously, and indeed intense downslope windstorms occur in both cases."

10- Figure 5: It is difficult to understand what is what cross section? Mention "up1" in the caption.

We agree with the reviewer. In the revised version we will indicate in the caption the meaning of UP1 and UP2. These points are the locations where we represented the upwind potential temperature profile (Figure 6).

11- Figure 9b,c: Easier to compare if you use the same domains? Did you discuss the missing northern stratiform cloud deck in the simulations?

We agree with the reviewer's consideration; it would be easier to compare Figure 9 b with Figure 9 c if we showed them in the same domain. However, it was not possible for us to obtain a clear satellite data representation for domain 05 only. The data used in Figure 9 is directly extracted and postprocessed from GOES-R raw database. We did our best to display the clearest possible image of the small lee cloud formation, and for this purpose we decided to zoom out this image so that the grey color scale is adjusted with the rest of the existent features (the other clouds, the lake, and the sea) and the lee clouds appear more sharply defined . In any case, in the revised version we will add another panel zooming in to exactly match domain 05 for easier comparison with the model results. Regarding the northern stratiform cloud deck

indicated by the reviewer, it is hinted by the model (you can see the edges of it in domain 5 in Fig 9a), but it is likely not correctly simulated because it is only very partially included in the domain and cloud water is not part of the boundary conditions for nested grids. We will mention this is the revised version.

Please also note the supplement to this comment:
https://www.earth-syst-dynam-discuss.net/esd-2019-3/esd-2019-3-AC2-supplement.pdf
* * *
2013-12-24T03:00:00

Fig. 1.

---

## Author Comment (AC3) · 23 May 2019

*Supplementary material uploaded: PDF with answers in an easy-readable format + figures. —- General: This paper reports on simulations of terrain induced wind systems in Mexico, and is novel in the respect that I have not seen much work in this particular region. The question that I have been wrestling with is however: "Is this novel science?"

The simulations are carried out for one single events, and while the authors may feels this was a typical event, the reader is not given much information other than some verbal arguments. So this remains a case study, and does not even claim to be general. The type of flows that are described (mainly downs-slope wind-storms and hydraulic

jumps) occur at many places around the world and has been studies extensively both from observations and with models. It is well known that numerical models perform well in these type of "hydraulic flows"; hence it does not reveal anything new about the flow features other than such that are directly related to specific local features. The authors do not even try to show that this type of flows have special characters because of the geographic location, which closer to the equator than in most other studies.

Hence, there is nothing new here and from that respect I should have recommended "reject". However, there is nothing fundamentally wrong with the study as such, so maybe that would be unfair. So I will leave to the editor to determine if the world needs one more paper about these events. I would tend to say "no", but in the case the editor say "yes" the paper still needs major revision. —- We thank the reviewer for accepting the evaluation of this manuscript; we appreciate the effort of his/her thorough review. The reviewer's general concern about the significance of our research to the existent science in the field is well taken; perhaps we have not made our contribution apparent enough in the text. We consider that the manuscript has several outstanding points that make it a noteworthy publishable work for this journal:

1) It is true that Tehuantepecers are orographically induced flows, of the kind occurring elsewhere on the planet. They are, however, of the very few with a global significance. Winds funneled through the Isthmus of Tehuantepec, are so intense and extend so far out to sea, that they are clearly visible from space, in scatterometer data, microwave derived total precipitable water, and even forming large rope clouds in their outflow/frontal boundary. Furthermore, Tehuantepecers produce strong upwelling in the Gulf of Tehuantepec resulting in large chlorophyll blooms that are critical for the food chain and marine life in the eastern Tropical Pacific. So, they are not just one more of this type of flows.

2) The scope of the paper is not to demonstrate anything particularly special about these orographic flows in terms of physics. Instead, our research's goal is to show that intense flow acceleration in the Isthmus of Tehuantepec during Tehuantepecer events

is not restricted to the well-known mountain gap wind jet off Chivela pass, but occurs in the neighbouring Sierras as well, in a stretch of over 100 km and for different dynamical mechanisms, forming downslope windstorms and hydraulic jumps. Our research objective is scientifically relevant because, while much attention has been given to the gap winds through Chivela pass, very little is known about the flow structure and associated extreme phenomena developing elsewhere across the Isthmus. Our work is the first, as far as we know, to analyze these extreme winds and discuss their driving dynamics.

3) Knowledge about these phenomena is also important for social and economic reasons because of the major impact that they have on the region. These extreme events cause, every year, problems and accidents involving population as well as infrastructure (please, see the references in the Introduction section), and a better understanding about them can help to mitigate the damages that they produce. Moreover, the region is the most important for wind energy generation in all of Latin America. These events directly affect the production of wind turbines and could undermine their performance if not considered in wind farm operations.

In the new version of the manuscript we will bring out more clearly the goal of our study and why it is novel science (point 2 above), building upon previous studies on Tehuantepecers.

Tehuantepecers are a recurrent feature of the circulation in the area, particularly in winter months, as we explicitly mention and support with references in the Introduction section. We simulated other events during late 2013 and also the previous winter season in early 2013, and we saw that downslope windstorms and hydraulic jumps also develop in the flow across the Isthmus. The case we present in the paper is the most intense and long lasting in the period, but we are very certain that the occurrence of these extreme phenomena is commonly associated with Tehuantepecer events in general. Evidences come not just from our model data, but also from wind company reports and from the damages they cause.

With regard to the use of a model for the study, it is of course not a novelty and we do not claim it to be so at all in the paper. It is precisely because models perform well in simulating these orographic flows, as the reviewer mentions, that we use one to analyze the flow structure and explain the particular mechanisms producing downslope windstorms and hydraulic jumps in the area. Unfortunately, there are no direct observations to help with the task, so a model is the best approach we can find. —- I have two main concerns:

1)he analysis if the model results is rather superficial and make many claims that are hard to substantiate from the graphic material presented. I'm sure this could be turned into a nice paper if the authors try to think a bit out of the box and perform an actual analysis of the model results, rather than just produce a few standard plots direct from the model data. Much of the theoretical background includes the planetary rotation; this is not commented on at all and here may be an opportunity to take a novel angle, comparing these results to higher-latitude cases. Just in general I would like to see a more in-depth analysis. —- The Rossby number for the flows in our study is high (above 13), given the small width of the mountains (about 35km in both cases), the strong wind speed of the order of 20m/s and the low latitude (f is about 4x10-5rad/sec, less than half the value in mid-latitudes). The role of planetary rotation is therefore very minor when compared with the effect of inertia, pressure and gravity forces. Mountain waves and related orographic flows such as those producing downslope windstorms and hydraulic jumps are in general high Rossby number motions everywhere, even at much higher latitudes; thus, in all related theoretical studies we know of (certainly in all those cited in the paper), the Coriolis force is neglected.

However, the effect of planetary rotation, or rather the lack of it, is indeed very relevant to explain the large extent over the ocean of the accelerated flow exiting the Isthmus. As mentioned above, the latter is instrumental to make Tehuantepecers stand out for their size and impacts among other orographic flows. Is it, perhaps, this effect what the reviewer's concern is about?
Gap winds and downslope windstorms in mid and high-latitudes do not usually extend too far downstream from the mountains where they develop; the existing accelerated flow adjusts quickly to the general geostrophic synoptic flow in the region. Even in gap winds through marine straits, where the exit region is free from topographic obstacles as in Tehuantepecers, the outflow weakens and merges with ambient circulations in relatively short distances. The article by Steenburgh et al (1998), cited several times in the text, discusses the mechanisms behind the large downstream extension of Tehuantepecers over the Gulf of Tehuantepec arguing that it is the combined effect of the lack of obstacles on the marine surface and the low f parameter of the tropical latitude of the area. The small Coriolis force results in very weak synoptic circulation (small pressure gradients and low wind speeds) and hence the gap wind does not encounter any ambient large-scale flow to merge with, describing a very-close-to-inertial trajectory. Furthermore, a small Coriolis force produces weak wind deflection and thus the anticyclonic curvature of the gap jet as it moves over the ocean is not very pronounced. These authors perform numerical experiments for the Tehuantepecer case they studied with the f of 45ïĆř N and show that in mid latitudes the gap wind outflow would curve much more westward, thereby not reaching as far south as in the actual situation.

Another result in the numerical experiments in Steenburgh et al. (1998) with different f values, is that the gap winds and flow in the Isthmus itself are largely unaffected by changes in the magnitude of the Coriolis parameter. Therefore, as theory predicts, planetary rotation plays only a minor or negligible role in the development and dynamics of these orographic circulations.

As per the reviewer's suggestion, we will comment on planetary rotation and the impact of the low latitude in Tehuantepecer's structure (mostly on the extent of the outflow in Gulf of Tehuantepec) in the Introduction section of the revised manuscript, and also when discussing the large reach of the accelerated flow resulting from the downslope windstorm in the lowest mountain in our analysis.

—- 2) The main results are all together expected and reveals nothing about this flow

that couldn't have guessed without the model simulations. Sure there are details here that no one could guess, but much of that detail is not harvested. —-

We disagree. We haven't seen any mention of downslope windstorms or hydraulic jumps developing in the Isthmus in any study related to Tehuantepecers, not even in the highly referenced article by Steenburgh et al. (1998), discussed above, detailing the dynamics of the gap winds and their outflow over the Gulf of Tehuantepec with the use of numerical simulations. If it was so evident that the flow shows the behavior we analyze in our work, it would have been reported in some of the several studies dealing with Tehuantepecers. The focus is always on the gap wind jet off Chivela pass. In Steenburgh et al. (1998) there is only a very brief comment about mountain waves and flow acceleration occurring also on the lee slope of the mountains east of Chivela pass, where their trajectory analysis shows that the cold air is also able to cross over to the Pacific side of the Isthmus, but with no further elaboration. The reason is likely related to the low resolution (6.67 km) or to the now outdated MM5 model used in this early study, which were not capable of simulating the downslope windstorm and hydraulic jump phenomena. Thus, this provides evidence that high-resolution simulations with an adequate tool (the WRF model) are indeed necessary to reveal these flow features. They are certainly not captured by global analysis and there are virtually no station observations or observational campaigns that can disclose their existence and structure. —- Finally, the language would benefit from editing by a native english speaker. —- We will ask a native speaker to edit the manuscript, as per the reviewer's suggestion.

Specific:

1- Make sure you define abbreviations the first time you use it, and stick with abbreviation afterwards. HJ is not explained, and is used interchanging with "hydraulic jump".

This is true, we will correct this in the revised version of the manuscript.

2- Drop the entire section 1.1; this is textbook stuff and just take up space.
Section 1.1 aims to put in context the hydraulic analog theory applying to the studied flows. We considered it important because it helps to better understand the discussions, as we often refer to this theory throughout the article. However, we understand the reviewer's concern, and in the revised manuscript we will reduce this section.

3- If you feel a need to validate the model, this should come before results, not after. Moreover, the cloud evaluation is superficial to the point of being useless. Either drop it or develop it.

The validation we are performing is not a general validation of the model, since there are very few observations (just two sites) and not exactly in the best locations in relation with the phenomena we are studying. To begin by validating d04 and d05 even before explaining what we are analyzing, could confuse the reader and make explanations and discussions difficult to follow. It seems more appropriate and natural to us to first analyze the model results and then verify that they indeed agree with point observations.

With regard to the cloud evaluation, we do not think it is useless at all. That lee wave clouds exist in the same location and with the same extent as in the model results, strongly suggests that the model solution is accurate in simulating trapped lee waves precisely in the focus area, and therefore realistic. In the revised version of the manuscript, we will show the cloud image and model results over the same exact domain, to make comparisons easier and highlight the value of this piece of evidence.

4- P3, l25: What do you mean by microscale?

We mean atmospheric motions of spatial scales less than 2km, following the definition of microscale in the American Meteorological Society (AMS) glossary of meteorology. The resolution of the innermost grid is 444m, sufficient to resolve some of these motions, including rotor circulations and even the hydraulic jump itself. However, since we are really referring to details of the downslope windstorms, while microscale meteorology is most often dealing with turbulence and other truly small-scale processes, we are

going to change the word microscale and rephrase the line:

"In section III, the primary results obtained are shown, divided by the synoptic-mesoscale situation, the upstream-downstream structure of the phenomena and the microscale situation" By: "In section III, the primary results obtained are shown, divided into synoptic-mesoscale situation and upstream-downstream structure of the phenomena"

The word microscale in the line the reviewer is referring to is actually correct, since the WRF model can run in L.E.S (large eddy simulation) mode resolving turbulent eddies, which are indeed microscale motions.

5- Figure 1: Why are the two most high-resolution domain off center wrt to the gap?

Precisely because the focus of our study is not the flow through the gap (Chivela pass) but in the mountains around, especially those to the east. Perhaps we didn't make it apparent enough in the article. We will modify the Introduction section, as mentioned above, to highlight more clearly the goal of our work, and what sets it apart from previous literature on Tehuantepecer winds.

6- P6, l3: How much of a spinup is required before the model physics is realistic?

From 3 to 6 hours is usually recommended. This is a standard practice in WRF simulation with this resolution [1]. Downslope windstorms and hydraulic jumps form around 12 UTC December 23, which is when we start the simulations. We show results from 3h into the simulation in Figure 5 only, to illustrate how the phenomena that we want to study develop. Most of the analysis is from data with a spinup time of 15h and more, when the flow features are fully mature. [1] Warner, T. T. (2011). Quality assurance in atmospheric modeling. Bulletin of the American Meteorological Society, 92(12), 1601-1610.

7- P6, l18: This is incorrect; U* is not a wind speed; its a scaling parameter that depends on the vertical turbulent momentum flux. Also explain how the logarithmic

wind-law is applied; with this formula, the wind just increases with height, so you need an anchor point.

We agree with the reviewer in that U* is not a real velocity but a parameter related to the vertical turbulent momentum flux. It is a reference velocity or a velocity scale, whose square value yields the magnitude of the vertical turbulent momentum flux near the surface, where it is assumed to be independent of height and nearly constant. It has dimensions of velocity (units are m/s) and it is called friction velocity in all text books and literature we know of. Perhaps the reviewer concern is related to the ws* symbol we used in the text, which can be confused with a real wind speed, such as wsz. We will change the ws* naming for this parameter in Equation 3 to the more standard U* symbol, so that no confusion can be made with an actual windspeed.

8- Figure 2: What is the height of the 850 hPa wrt mountain crest? Why pressure levels at all, and not model levels? Note the warmest temperatures hanging on the southward facing lee slope; DSWS already happen here? Maybe the D01 domain is a bit small, or could have been located farther north, since there's a lot of uninteresting ocean south of the coast.

Figure 2 displays the synoptic setting for the Tehuantepecer event in our study. It is not depicting model results, but global analysis data at 25km resolution. The use of pressure levels is a standard practice for this type of plots. We show 850 hPa temperatures and surface pressure because we are interested in the situation at low levels. The purpose of the figure is to illustrate how the low-level cold air driven by cold air damming in the Rockies continuing in the Sierra Madre Oriental in Mexico, moves fast southward, reaching the Bay of Campeche. The 850hPa surface is around 1600m in Mexico, within the cold air mass, which tops at about 2500m (see the sounding in Fig 3d). The tallest mountain crests in the Isthmus are around 2000m.

We do not see any sign of DSWs in these images; there is not enough detail, and as the reviewer suggests, a map in pressure levels might not be the most appropriate for

the task. The D01 domain is centered in the Isthmus. It includes a significant marine portion in the south because of large extent of the Tehuantepecer outflow into the ocean, as the reviewer can observe in Figure 3a. It is convenient to set the boundary relatively far downstream from the area of interest to avoid numerical problems related to the imposed lateral boundary conditions, such as wave reflections that can perturb the solution within the domain.

9- Figure 3: Results are impressive but not unexpected. Moreover, legend for panel (d) is missing.

We are glad that the reviewer finds these results impressive. They might be expected for someone with the level of expertise of the reviewer, but we believe that the general audience of the journal and especially those with interests in the area will find them revealing. The missing legend for panel (d) will be included in the revised version. Thank you for noticing.

10- P8, l8: What is the Rossby radius of deformation here.

If we consider the depth of the surge to be around 2500m and the Brunt Väisälä frequency 0.012 s-1, the Rossby radius of deformation is approximately 750 km. For comparison, the Isthmus is about 200 km across.

Clarke (1988) argued that since pressure gradients are so weak over the Gulf of Tehuantepec, the wind should follow a close to inertial trajectory. The radius of this inertial path for an outflow velocity of 25 m/s and 15N latitude is 662 km, which appears to be in the range of what we see in our simulations, at least along the main outflow axis, where the cross-flow pressure gradient is very small. Steenburgh et al (1988) found this to also be the case, as they discuss in detail the balance of forces for the gap outflow over the Gulf of Tehuantepec, explaining its fan like structure and curvature. In our work, as noted before, the focus is not on the gap outflow, but on the smaller scale extreme wind phenomena occurring in the Isthmus. We will, nevertheless, mention the reasons for the outflow's shape in the introduction section when

briefly commenting on the effect of planetary rotation (see response to main concern 1, above). Clarke, A. J., 1988: Inertial wind path and sea surface temperature patterns near the Gulf of Tehuantepec and Gulf of Papagayo. J. Geophys. Res., 93, 15 491–15 501

11- P8, L9: Are you here referring to the cross-over in the T-shaped structure? Might this is the DSWS? There seems to be a tremendous along-flow convergence/divergence here.

Yes, exactly, we are referring to the strong flow acceleration represented in Figure 3b, with a T-shaped structure. As we show in detail later on, this is where the DSW occurs.

12- P9, l2: Here you argue that this is a "flow thinning" event and not a gravity-wave breaking downs-slope flow event, but later tit is the opposite.

The flow thinning we are referring to here is that occurring under the dividing streamline generated by gravity-wave breaking to the lee of the mountain. We will remove "flow thinning" from this sentence to avoid confusion with the one happening for example in the case where there is an inversion layer close to the mountain top.

13- P9, l4: And where is NP? Moreover, this analysis (d-panel) would have been much more informative if you had analyzed the depth of the cold air and plotted it as a contour plot, showing its geographical distribution

We agree with the reviewer's suggestion, in the revised version we correct Figure 3, better explaining where NP point is and introducing the caption for Figure 3d. Figure 3d contains more information than just the depth of the cold pool, which is relatively homogenous in the area, as seen for example in Figures 6b and 6c. It shows the stability of the column and the vertical wind profile. We prefer to maintain the current panels in Figure 3, but we will add a comment about the depth of the cold air in position NP being similar to that closer to the mountains.

14- P9, l5&6: If you want to use this type of sounding plot, you need to tell the reader

what's on the axes. If the gray transverse lines are isotherms, there is hardly any inversions at all; to me the lower one looks more like an isothermal layer while the upper (subsidence) may be a weak one.

The plot is a skew-T log-P diagram, very commonly used to represent upper-level soundings and quite standard in weather analysis. Perhaps it may be confusing that it is lacking wet and dry adiabats? We will add the dry ones, which are relevant for the discussions. We will specify it is one of such diagrams in the text and add further details about the barbs in the wind profile and their scale, which is missing in the caption.

The reviewer is right in that the inversions are fairly weak, especially the lower one. We will replace the wording "shows the depth of the cold air pool, defined by the inversion existent at about 800hPa or 2500m in the temperature profile" by "shows in the temperature profile a stable lower boundary layer capped by a very stable isothermal layer from 850hPa up to about 800hPa, or 2500m, defining the depth of the cold air pool"

We will also change the word "inversion" by "very stable isothermal layer" in other instances of the text discussing the depth of the cold air pool.

15- P9, l19: Awkward; what do you mean by "far reaching"?

We mean "reaching out as far as the mountain gap winds do". In the revised version of the manuscript we will rewrite this sentence to make it clearer for the reader.

16- Figure 4: Show the modeled wind speeds already here and save a plot later

With this figure, we want to show the whole Tehuantepecer event duration (from the 23rd of the December to the 29th), as reflected by the observations. We also show some days before and after to contrast Tehuantepecer winds with the typical flow regime in the area. Furthermore, the figure provides justification on the period chosen for analysis as the 36h interval of maximum intensity (highlighted in both observational timeseries). We consider that including model results here for the purpose of validation would distract the reader by making the figure rather messy, so we prefer to leave it as

it is, and keep the validation in a separate figure for a shorter interval allowing much more detail.

17- Section 3.2: drop first sentence; we just read about that, no need to repeat.

We agree we reviewer's comment, in the revised version of the article we will drop this sentence.

18- P10, l67; Awkward English; what is a "wind path"?

We mean "encompassing the flow path before and after crossing the mountains", we will correct this in the revised manuscript.

19- P10, line 19; "steeper" than what?

We wanted to say: 'asymmetric mountain with steeper lee than windward side', we will change this in the revised version.

20- P10, l23-24 and elsewhere: I don't dispute the wave-breaking argument, but how can you see this here? There are no temperature-gradient reversals that I can see, nor is there any TKE aloft that would result from it. I would have expected to see at least truly vertical isotherms and elevated layers of TKE, or gradient reversals and no TKE.

Vertical isentropes are more clearly seen at early stages and in the case of the higher mountain (see Fig 5d), but they are also present in Fig 5a for the lower mountain. They occur in the area that appears dark blue later on (Fig 5g and 5j) showing very low or close to zero windspeeds, and where isentropes are split apart from each other, indicating a well-mixed region. Overturning isentropes and temperature-gradient reversals are observable in the case of the higher mountain only, in Fig 5d. We will include as supplementary material an animation of figures like those in Fig 5 to show the process of development of the DSWs. In the animated sequence wave breaking is more apparent.

We agree with the reviewer in that there should be higher TKE values in the same layers

where wave breaking occurs, but this variable is the result of the PBL parameterization in the model and thus it only considers subgrid-scale variations in wind speed due to turbulent eddies, as represented in the scheme. The turbulence due to wind shear in the column is well captured; however, the turbulence associated with gravity waves, due to rotors and non-local turbulence advection is not represented and accounted for, because a much higher horizontal resolution of the order of tens of meters would be needed. This problem in numerical models is well known (see for example the review paper by Vosper et al. 2018) and we will introduce a comment about it in the revised version of the manuscript.

Simon B. Vosper, Andrew N. Ross, Ian A. Renfrew, Peter Sheridan, Andrew D. Elvidge and Vanda GrubisÌŇic ÌĄ, 2018: Current Challenges in Orographic Flow Dynamics: Turbulent Exchange Due to Low-Level Gravity-Wave Processes. Atmosphere, 9, 361. doi:10.3390/atmos9090361

21- P10, l28: What do you mean by "bounded by turbulence"?

We are describing Figure 5i disposition of TKE highest values, in shaded green. We mean "confined by layers of strong wind shear and turbulence", we will reword the sentence in the revised text.

22- P10, l31 The use of Fr is a powerful but yet blunt instrument to analyze these flows. I have two concerns here: 1) As the air has propagated up til crest of the terrain, Fr is already modified. The classical analysis by Durran, cited earlier, also uses the upstream Fr before the flow has hit the terrain; not that at the top of the hill. Hence I would have liked to see the truly upstream Fr instead; not the value that has already been modified. 2) Is it certain that the air reaching the observation stations actually comes from the point directly north of the station? A trajectory analysis would clarify the 3D dynamics of the flow.

FIGURE1 ATTACHED BELOW

The figure above shows the Froude number at upstream points 1 (low mountain) and 2 (high mountain). The values are not so different from those shown in the paper, with Fr close to 1 upstream of the high mountain where the strong HJ forms and around 2 in the case of the low mountain. We will now include the calculation of Fr upstream instead of at the top of the mountain as per the reviewer's suggestion.

With regard to the question about the upstream trajectory of the air reaching the observation stations, we do not fully understand the reviewer's concern. The wind is from the north and quite steadily at low levels, so we consider that the cross sections in the north-south direction serve very well the purpose of showing the vertical structure of the flow as it crosses the mountains.

23- Figure 5: To much information in too too many too small panels. In fact, you could easily get rid of one third, by plotting the TKE and w in the same panels, as you do with temperature and winds.

Because of the relation between TKE and wind shear we would prefer to keep the panels showing both TKE and wind speed contours. Adding vertical velocity as an extra layer would make them very difficult to read. We think that is best to keep Fig 5 as it is, unless the reviewer has a very strong objection. With its high resolution, the figure can be readily expanded to reveal all fine scale details very clearly.

24- P11, l1: The position of the hydraulic jump, which is far from very distinct to begin with, hardly makes any propagation clear. Instead, analyze the position of the jump at different times and plot the position, and maybe also its strength, as a function of time. Maybe also as a function of Fr. From this the reader cannot really see any propagation.

The small hydraulic jump lies around latitude 16.40 in Fig5a, but it is only easy to spot if one sees the sequence in motion. We will now point at the animation in the supplementary material (see response to comment 20 above) instead of Fig5a alone when commenting on this modest hydraulic jump.

[Figure]

25- P12, line 1 Use "indicating" instead of "signalling"

We agree with reviewer's suggestion. We will use "indicating" in the revised version of the article.

26- P12, l5; "further high" is awkward.

The sentence "Stability is much reduced further high,..." will be rewritten as "Stability is sharply reduced aloft..."

27- P12, l9: Can't see any wave overturning in these plots. This may be because mixing erodes overturning isotherms before they can be seen in the model output. In that case, there should be TKE there, which I can't see either

FIGURE2 ATTACHED BELOW

Fig 5 d from the paper is shown above with a white oval highlighting isotherm overturning, clearly visible for theta = 308 K and theta = 310 K to the lee of the mountain. We will add a similar marking in the revised manuscript.

The reviewer is right in that mixing erodes isotherm overturning very quickly, both with resolved vertical motions responding to the created instability and by subgrid turbulent eddies from the PBL scheme. The TKE variable is only reflecting the latter contribution; therefore, it has a smaller value than it should. In addition, the PBL scheme is more designed to account for convective instability and turbulent eddies rising from the surface due to heating, than to deal with instability resulting from gravity wave overturning, like what we see here. Thus, it is also underestimating TKE in this case. As mentioned in the response to comment 20, this is still an open issue in numerical modeling. We will add a comment about it in the revised version of the manuscript.

28- P12, l30: Awkward; "slightly ... d05".

We agree with the reviewer. In the revised version, we replace the sentence "...slightly better so those from d05,..." by "...slightly better in d05 case ..."

29- P12, l34: Also awkward; "which situations".

We will replace "which registers HJ situations" by " downwind from the strong HJ"

30- Conclusions contain far-reaching statements that cannot be substantiated by one case study

We will tone down some of the concluding statements. In particular, we will remove the ending sentence "It is likely that the depth of the cold pool and how it compares with topographic barrier height, is a key factor determining the extent, location and intensity of these lee wave phenomena and whether they take place at all", which we agree it is unsubstantiated from just one case study.

Please also note the supplement to this comment:
https://www.earth-syst-dynam-discuss.net/esd-2019-3/esd-2019-3-AC3-supplement.pdf

[Figure]

UPSTREAM1 and UPSTREAM2

**Fig. 1.**

[Figure]

2013-12-23 15:30 UTC

2013-12-24 03:00 UTC

**Fig. 2.**